# Moesin (*MSN*) as a Novel Proteome-Based Diagnostic Marker for Early Detection of Invasive Bladder Urothelial Carcinoma in Liquid-Based Cytology

**DOI:** 10.3390/cancers12041018

**Published:** 2020-04-21

**Authors:** Jeong Hwan Park, Cheol Lee, Dohyun Han, Jae Seok Lee, Kyung Min Lee, Min Ji Song, Kwangsoo Kim, Heonyi Lee, Kyung Chul Moon, Youngsoo Kim, Minsun Jung, Ji Hye Moon, Hyebin Lee, Han Suk Ryu

**Affiliations:** 1Department of Pathology, Seoul National University College of Medicine, Seoul 03080, Korea; hopemd@hanmail.net (J.H.P.); fejhh@hanmail.net (C.L.); blue7270@gmail.com (K.C.M.); jjunglammy@gmail.com (M.J.); sssince2004@naver.com (J.H.M.); 2Department of Pathology, SMG-SNU Boramae Medical Center, Seoul 07061, Korea; 3Department of Pathology, Seoul National University Hospital, Seoul 03080, Korea; minji891111@nate.com; 4Division of Clinical Bioinformatics, Biomedical Research Institute, Seoul National University Hospital, Seoul 03080, Korea; hdh03@snu.ac.kr (D.H.); kksoo716@gmail.com (K.K.); hylee4161@gmail.com (H.L.); 5Proteomics Core Facility, Biomedical Research Institute, Seoul National University Hospital, Seoul 03080, Korea; 6Department of Pathology, Samsung Changwon Hospital, Sungkyunkwan University School of Medicine, Changwon 51353, Korea; hymedljs@naver.com; 7Center for Medical Innovation, Biomedical Research Institute, Seoul National University Hospital, Seoul 03082, Korea; km601@naver.com; 8Department of Biomedical Sciences, Seoul National University College of Medicine, Seoul 03080, Korea; biolab@snu.ac.kr; 9Department of Radiation Oncology, Kangbuk Samsung Hospital, Sungkyunkwan University School of Medicine, Seoul, 03181, Korea

**Keywords:** bladder urothelial carcinoma, invasion, biomarker, proteomics, liquid-based cytology, moesin

## Abstract

Bladder urothelial carcinoma (BUC) is the most lethal malignancy of the urinary tract. Treatment for the disease highly depends on the invasiveness of cancer cells. Therefore, a predictive biomarker needs to be identified for invasive BUC. In this study, we employed proteomics methods on urine liquid-based cytology (LBC) samples and a BUC cell line library to determine a novel predictive biomarker for invasive BUC. Furthermore, an in vitro three-dimensional (3D) invasion study for biological significance and diagnostic validation through immunocytochemistry (ICC) were also performed. The proteomic analysis suggested moesin (*MSN*) as a potential biomarker to predict the invasiveness of BUC. The in vitro 3D invasion study showed that inhibition of *MSN* significantly decreased invasiveness in BUC cell lines. Further validation using ICC ultimately confirmed moesin (*MSN*) as a potential biomarker to predict the invasiveness of BUC (*p* = 0.023). In conclusion, we suggest moesin as a potential diagnostic marker for early detection of BUC with invasion in LBC and as a potential therapeutic target.

## 1. Introduction

Bladder urothelial carcinoma (BUC) is the most common and lethal malignancy of the urinary tract [1]. The evaluation of cancer cell invasion beyond the subepithelial layer is crucial due to the availability of different therapeutic approaches [2], which has prompted the identification of robust invasion-associated molecular characteristics of BUC [3,4].

Unlike extensive genome-based studies, in-depth proteomic analysis of BUC has been introduced in a few studies [5,6,7,8,9,10], among which only one study identified predictive markers for tumor invasion in muscle-invasive BUC (MIBUC) by predominantly using human-derived tissue samples and proteomic approaches [7].

In our previous study, we successfully conducted a proteomic study and validated a proteome-based novel diagnostic marker of BUC in liquid-based cytology (LBC) [11], which is the gold standard test for surveillance of urothelial carcinoma recurrence or progression [12,13]. Voided urine cytology used in this study is the standard non-invasive method for the detection of BUC by the assessment of morphologic changes of exfoliated urothelial cells in comparison to transurethral endoscopy which is more invasive and expensive. Recent cytologic slide-based ancillary tests showed anticipatory positive results by direct integration with cytomorphologic findings [14].

Moesin is a protein encoded by the *MSN* gene at chromosome location Xq12 as a member of the ERM (ezrin, radixin, and moesin) family [15,16] and is known to be associated with an aggressive phenotype in several malignant tumors [16,17,18,19]. Higher *MSN* mRNA expression was also significantly associated with unfavorable survival in various types of human cancers, including lung, stomach, and pancreatic cancer in The Cancer Genome Atlas (TCGA) dataset [20].

In this study, we explored proteome-based novel biomarkers to predict advanced tumor stage in voided urine cytology samples collected by liquid-based preparation and evaluated the predictive ability of moesin (*MSN*) in the context of BUC invasion through immunocytochemistry (ICC) validation in independent LBC cohorts. Finally, we investigated its functional role in cancer invasion with a three-dimensional (3D) in vitro invasion assay.

## 2. Results

### 2.1. Proteomic Analysis Identified Cancer Invasion-Associated Protein Groups in Urine Liquid-Based Cytology

In total, 3259 and 1779 proteins were identified and quantified at the 1% false discovery rate (FDR) level by a single-shot proteomic analysis of the LBC samples (Appendix A). For each case, the average number of identified and quantified proteins was demonstrated in Appendix A. Multigroup tests with one-way ANOVA revealed 182 differentially expressed proteins (DEPs) among three cohorts based on pathologic T (pT) stage (Figure 1A, Appendix A). The hierarchical clustering was classified into four groups based on the proteome expression (Figure 1A). The protein expression of *KPNA3* [21], *TACSTD2* [22], *GRHL2* [23], *NCAM2* [24], *LLGL2* [25], *ATP1B1* [26], *CLTC* [27], *PARP4* [28], *MVP* [29], and *PPA2* [20] that play a tumor-suppressive role was increased in non-invasive BUC (NIBUC) compared to invasive BUC (Group 1). On the other hand, several proteins promoting cell motility and invasion, including *ARHGEF2* [30], *MSN* [16,18,19], *VIM* [31], *LCP1* [32], *FLNA* [33], *FERMT3* [34], *ITGAM* [35], and *CORO1A* [36] were significantly upregulated in MIBUC (Group 3, Figure 1A). A further two-group analysis between NIBUC and MIBUC also demonstrated the overexpression of DEPs with a tumor-suppressive role, including *LGALS3* [37] and *VAPA* [21] in NIBUC (Figure 1B, Appendix A). Several key proteins such as *MAP2K1* [38], *ITGB4* [39], *ITGA6* [40], *PTPN6* [41], *FMNL1* [42], *ANXA1* [43], and *MMP9* [44] that modulate cell motility and tumor cell invasion were upregulated in MIBUC (Figure 1B, Appendix A). Together, our proteomic findings suggested a cooperative interaction among several genes in the invasive process of BUC.

Subsequently, a gene ontology analysis on biological process revealed enrichment in cytoskeleton organization, cell migration, and cell motility, which implicated significant alterations in the cytoskeletal architecture and invasion process (Figure 1C, Appendix A). Especially, DEPs involved in cell motility and invasion were mostly upregulated in MIBUC compared to NIBUC. A further comparison of stromal-invasive BUC (SIBUC) and NIBUC revealed that biological processes with ribonucleoprotein complex biogenesis and antigen processing/presentation of peptide antigen were significantly enriched in SIBUC by upregulated and downregulated DEPs, respectively (Appendix A). Molecular functions with UFM1 activating enzyme activity and oxidoreductase activity were enriched while comparing MIBUC and SIBUC groups.

### 2.2. Proteomic Library of BUC Cell Lines Identified Candidate Biomarkers

For the discovery of candidate biomarkers related to invasion, we performed a tandem mass tag (TMT) proteomic analysis and constructed a BUC cell line proteomic library (Figure 2, Appendix A). First, we assessed the invasion and migration ability of eight BUC cell lines to categorize them into invasive BUC cell line (IBUC_CL) and non-invasive BUC cell line (NIBUC_CL). Among the BUC cell lines, T24, J82, and 253J-BV (IBUC_CL) revealed the most invasive and proliferative capacity, while RT4, HT1376, and HT1197 showed the least aggressive ability (NIBUC_CL) (Figure 2A,B). Next, we conducted a proteomic analysis between IBUC_CL and NIBUC_CL for the discovery of candidate biomarkers related to cancer invasion and identified 677 DEPs and aforementioned proteins in LBC proteomics, including *ATP1B1*, *CLTC*, *GRHL2*, *KPNA3*, *LDHB*, *LLGL2*, *MSN*, *MVP*, *NCAM2*, *PARP4*, *PPA2*, and *VAPA* (Figure 2C, Appendix A).

### 2.3. Multi-Omic Platforms Selected Moesin (MSN) as a Potential Biomarker for Invasive BUC

For the discovery of potential biomarkers related to invasion, we performed a stepwise analysis on a multilayer platform (Figure 3A, Appendix A). First, we compared DEPs from one-way ANOVA and paired t-test of LBC samples. Proteomic analysis of LBC revealed 182 DEPs and 188 DEPs in one-way ANOVA and paired t-test, respectively. Cross-validation with these two platforms showed 139 common DEPs in LBC proteomics. Next, we employed an extra platform of BUC cell line proteomics to determine a predictive biomarker for bladder cancer invasion. We used the 677 proteins for BUC cell line proteomics for a comparative analysis with DEPs derived from human LBC proteomics. Consequently, the cross-validation of DEPs from LBC and BUC cell lines revealed 12 invasion-associated biomarker candidates: *ATP1B1*, *CLTC*, *GRHL2*, *KPNA3*, *LDHB*, *LLGL2*, *MSN*, *MVP*, *NCAM2*, *PARP4*, *PPA2*, and *VAPA*.

To shortlist the optimal candidates, we evaluated the change in proteomic intensities among groups based on cancer invasion in all the proteomic data, including that of label-free LBC and the cell line TMT (Figure 3B, Appendix A). Seven out of the 12 candidates showed a random alteration of protein intensity regardless of the advanced tumor stage in BUC patients and of invasive phenotype in cell lines, which were eventually excluded for further validation tests. The remaining five candidate biomarkers, namely, *GRHL2*, *LLGL2*, *MSN*, *NCAM2*, and *VAPA*, demonstrated a gradual increase of protein intensity in groups with more invasive phenotypes, which guided us to select them as the final candidates for further validation.

In gene ontology analysis, the five final candidate biomarkers were associated with cellular transport (*LLGL2* and *VAPA*), immune response (*VAPA*), invasion process (*LLGL2* and *MSN*), and cellular organization (*GRHL2* and *MSN*) (Figure 3C). All four candidates, except *MSN*, were downregulated in MIBUC. *GRHL2* affects cell morphogenesis and epithelial–mesenchymal transition (EMT) and acts as a tumor suppressor in various tumors [22,45]. *LLGL2* and *VAPA* are involved in cellular transport and affect invasion process. These genes show a tumor-suppressive role in various tumors [21,24]. On the other hand, *MSN* expression was upregulated in MIBUC and consistent with its oncogenic role in invasion process [16,18,19].

### 2.4. The Inhibitory Effect of Moesin (MSN) Depletion on Cancer Invasion in BUC

Next, we performed a two-dimensional (2D) invasion and migration assay with T24 and J82 BUC cell lines to evaluate how the five selected candidates modulated the invasion ability of BUC cells (Figure 4A). The invasion and migration assay showed that BUC cells were significantly reduced in both the *MSN*-depleted cell lines as opposed to other candidate biomarkers including *LLGL2*, *NCAM2*, and *VAPA* that all failed to prove significant alteration of invasion ability (Appendix A). A further 3D invasion assay showed concordant findings that *MSN* knockdown T24 and J82 BUC cells exhibited a remarkable reduction in cell invasion (Figure 4B,C). A further pre-ranked gene enrichment analysis utilizing gene ontology term-defined gene sets linked *MSN* to proteins involved in actin dynamics (*CORO1A*, *FLNA*, and *LCP1*), formin (*FMNL1*), integrin signaling (*FERMT3*, *ITGAM*, *ITGA6*, and *ITGB4*), extracellular matrix (ECM) remodeling (*MMP9*), EMT phenotype (*VIM*), small GTPase activator (*ARHGEF2*), and mitogen-activated protein kinase (MAPK) pathway (*MAP2K1*) that were upregulated in the MIBUC group (Figure 5). A further co-expression analysis using TCGA data revealed strong correlation of *MSN* expression with proteins involved in actin dynamics (*FLNA*), integrin signaling (*ITGAM*), and EMT phenotype (*VIM*) and suggested a co-operative role of *MSN* with signaling pathways associated with cell motility (Appendix A).

### 2.5. Slide-Based Moesin Immunocytochemical Test Predicts Invasive Urothelial Carcinoma on Urine Liquid-Based Cytology

We further verified the diagnostic role of moesin to predict BUC invasion and its clinical application through ICC based on an independent urine LBC cohort, which was composed of NIBUC, SIBUC, and MIBUC. The proportion of moesin immunoreactivity significantly increased with BUC invasion—38.5%, 80.0%, and 85.7% in NIBUC, SIBUC, and MIBUC, respectively (*p*-value = 0.046; Figure 6, Table 1). The predictive ability was more powerful in a dichotomous comparison between the BUC group without invasion and the other group with invasion (*p*-value = 0.023, moesin immunoreactive rates, 38.5% vs. 82.4%, respectively).

## 3. Discussion

In this study, we prioritized moesin (*MSN*) as a protein biomarker for early detection of BUC invasion using a liquid-based cytologic test, which is the most widely used clinical screening method for monitoring bladder cancer progression. Moesin showed predictive ability for invasion of BUC in the independent ICC cohort. In situ immunoreactivities of moesin on LBC slide-based tests revealed statistical discriminative power when more than one cell was immunostained in LBC slides, which was concordant with the previous study where positive immunostaining in any cancer cell was significantly associated with poor overall survival in BUC [17]. In the TCGA public dataset, the higher expression of *MSN* transcript was also marginally associated with unfavorable clinical outcomes (*p*-value = 0.061; Appendix A). The higher expression of *MSN* was also associated with advanced American Joint Committee on Cancer (AJCC) staging (*p*-value = 0.001) and angiolymphatic invasion (*p*-value = 0.050) (Appendix A).

Growing evidence has shown that moesin (*MSN*) plays a crucial role in invasion by cytoskeletal reorganization and EMT in various malignant tumors [16,18,19]. Although the functional relevance of *MSN* has not been fully revealed in urothelial carcinoma [9,17], our in vitro 3D spheroid invasion assay along with the 2D invasion assay confirmed significantly decreased invasion ability in *MSN*-depleted BUC cell lines. The 3D tumor spheroid invasion assay has advantages, for example, tumor spheroids mimic a more physiologic tissue-like morphology and recapitulate tumor cells and microenvironment [46,47]

Our proteomic data demonstrated that moesin (*MSN*) upregulation is one of the major factors for BUC invasion, which can be more critical as the previous proteomic analysis of urine extracellular vesicles revealed moesin as one of the candidate biomarkers for bladder cancer diagnosis [9]. In a further network analysis with protein–protein interactions, we confirmed a tight clustering of *MSN* with several key proteins, including *ITGA6*, *ITGB4*, *FERMT3*, *FLNA*, *LCP1*, *CORO1A*, *FMNL1*, *ARHGEF2*, and *MMP9*, all of which modulate membrane ruffling, lamellipodia and filopodia formation, cell–ECM interaction, and ECM remodeling that play a crucial role in cancer cell invasion [48] (Appendix A). Moesin binds to phosphatidylinositol 4,5-bisphosphate (PI(4,5)P_2_), CD44, and Na^+^/H^+^ exchanger 1 (NHE-1), which are all key factors for cytoskeletal reorganization by modulating integrin signaling and integrin complex formation [16]. The complex that consists of integrin subunit α6 (*ITGA6*) and subunit β4 (*ITGB4*) and sequentially interacts with laminin [49] and kindlin-3 (*FERMT3*) is involved in tumorigenesis by modulating tumor cell–ECM interaction [50]. Filamin A (*FLNA*), plastin-2 (*LCP1*), coronin-1A (*CORO1A*), and formin-like-1 (*FMNL1*) also affect cytoskeletal dynamics by modulating actin filaments which eventually prompt cancer mobility and invasion [51,52]. Additional molecular studies need to be carried out, focusing on the above-selected proteomic markers. The results and how they can be interpreted in perspective of previous studies and working hypotheses should be discussed. The findings and their implications should be discussed in the broadest context possible. Future research directions may also be highlighted.

## 4. Materials and Methods

### 4.1. Patient Selection and Clinicopathologic Review

A total of 16 surgically confirmed LBC samples and an independent BUC cohort of 30 LBC specimens encompassing NIBUC, SIBUC, and MIBUC were employed for quantitative proteomic analysis and verification of diagnostic utility of ICC, respectively (Table 2). This study was approved by the Institutional Review Board of Seoul National University Hospital (IRB No. H-1602-150-747). Detailed information can be found in the Appendix A.

### 4.2. Proteomics Analysis and Data Processing for Peptide Identification

Appendix A indicates the key steps in our approach for the proteomic discovery of novel biomarkers. Tumor cells from LBC slides were scraped and the peptide was digested using the filter-aided sample preparation (FASP) procedure as previously described [53]. Each sample was desalted [54] and was followed by LC-MS/MS analysis. For BUC cell lines, a proteomic analysis was performed after eight BUC cell lines, namely, T24, J82, 253J-BV, 253J, 5637, RT4, HT1376, and HT1197 (ATCC; Manassas, VA, USA), were categorized as IBUC_CL and NIBUC_CL based on their invasion and migration capacities. Each sample was labeled by TMT and was followed by LC-MS/MS analysis. The LC-MS/MS analysis was conducted using a Q Exactive Plus Hybrid Quadrupole-Orbitrap mass spectrometer (Thermo Fisher Scientific Inc., Waltham, MA, USA) and an Ultimate 3000 RSLC system (Dionex, Sunnyvale, CA, USA) as previously described [53,55]. MaxQuant version 1.5.3.1 (Max Planck Institute of Biochemistry, Munich, Germany) [56] with the Andromeda search engine [57] and Proteome Discoverer 2.1 software (Thermo Fisher Scientific Inc., Waltham, MA, USA) [58] with the SEQUEST-HT search engine were employed for processing LBC and cell line data, respectively. More detailed information is available in the Appendix A.

### 4.3. Cell Migration and Invasion Assays with Small Interfering RNA (siRNA) Transfection

The T24 and J82 BUC cell lines were selected to evaluate cell migration and invasion abilities. RNA interference siRNAs targeting *GRHL2*, *LLGL2*, *MSN*, *NCAM2*, and *VAPA* were employed, followed by transfection to BUC cells. Detailed information is available in the Appendix A.

### 4.4. Tumor Spheroids and 3D Spheroid Invasion Assay

Tumor spheroids were generated for suspension culture. Mixed collagen/Matrigel matrices were constructed as previously described [59]. The dissemination of spheroids was assessed under a phase-contrast microscope. A confocal laser scanning microscope (Leica TCS SP8; Leica microsystems, Wetzlar, Germany) was employed for the detection of stained F-actin. Phalloidin–rhodamine (Thermo Fisher Scientific Inc., Waltham, MA, USA; 1:100 in phosphate-buffered saline (PBS)) was used for visualization of the actin cytoskeleton in 3D spheroid cells. Appendix A contain additional information.

### 4.5. Immunocytochemical Analysis

Immunocytochemical staining was conducted on LBC slides. Immunostaining of moesin was performed using Benchmark XT (Ventana Medical System, Inc., Tucson, AZ, USA). A monoclonal mouse anti-moesin antibody (Santa Cruz Biotechnology, Dallas, TX, USA) was diluted to 1:500. The binding of the primary antibody was identified using an Optiview universal DAB kit (Ventana Medical Systems, Inc., Tucson, AZ, USA) according to the manufacturer’s protocol. ICC analysis defined negative expression for tumor cells with no moesin expression as opposed to positive expression when at least more than one tumor cell expressed moesin [17]. We also assessed the intensity and proportion of positive BUC cells for H-score evaluation [60].

### 4.6. Statistical Analyses

All proteomic datasets were submitted to the ProteomeXchange Consortium (http://proteomecentral.proteomechange.org) (project ID: PXD016437) [61]. ToppGene Suite resources (https://toppgene.cchmc.org/) [62] and String [63] were used for gene ontology annotation and interaction network model construction, respectively. Cytoscape version 3.7.1 (Institute for Systems Biology, Seattle, WA, USA) [64] was used for the illustration of the network model. Statistical analyses were conducted using the Perseus software (Max Planck Institute of Biochemistry, Munich, Germany) [65] for proteomic data. The H-score for ICC validation was analyzed by utilizing the Kruskal–Wallis test and Mann–Whitney *U* test for the comparison of BUC groups with the GraphPad Prism 8.0 program (GraphPad Software, Inc., CA, USA). The cross-tabulation analysis was conducted by Pearson’s χ^2^ test and Fisher exact test with IBM SPSS Statistics version 20 (IBM Corp., Armonk, NY, USA). Detailed information is available in Appendix A.

## 5. Conclusions

Taking advantage of advanced proteomic techniques, the present study identified a novel promising diagnostic biomarker that can be applied as a new ancillary test for the prediction of BUC with invasion using voided urine LBC samples, the most frequently used diagnostic sample in routine practice. We successfully demonstrated that the immunoreactivity of moesin can be utilized as a diagnostic marker for early surgical intervention. Further investigation will be necessary for our future studies to validate the predictive ability of moesin in a larger cohort.

## Figures and Tables

**Figure 1 cancers-12-01018-f001:**
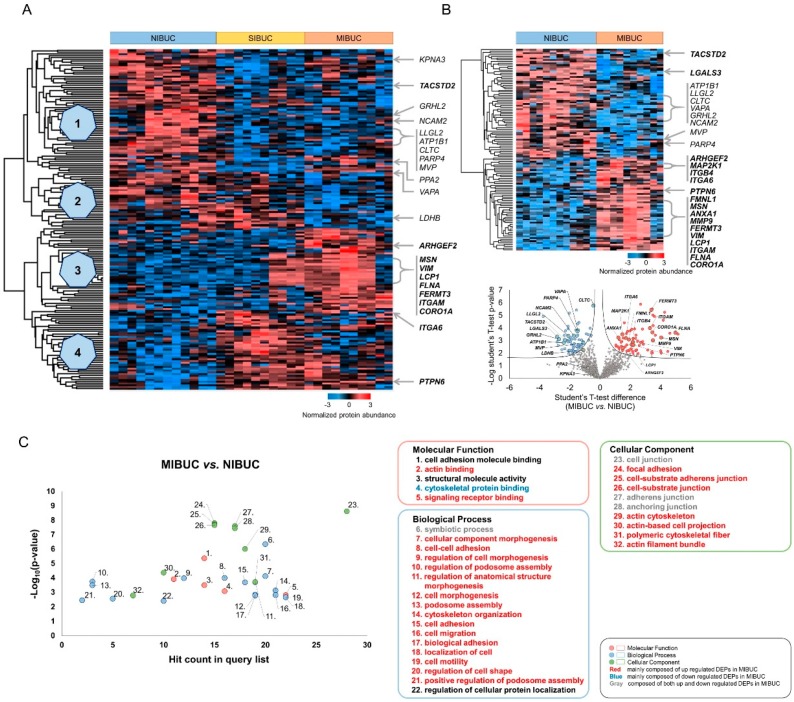
Results of proteomic analysis of bladder urothelial carcinoma (BUC) in liquid-based cytology (LBC) samples. (**A**) Hierarchical clustering of 16 BUC LBC proteomic data among non-invasive BUC (NIBUC), stromal-invasive BUC (SIBUC), and muscle-invasive BUC (MIBUC) (Group 1, downregulated in invasive BUC; Group 2, downregulated in MIBUC; Group 3, upregulated in MIBUC; Group 4, upregulated in invasive BUC). (**B**) Hierarchical clustering and volcano plot between MIBUC and NIBUC. (**C**) Gene ontology results between MIBUC and NIBUC.

**Figure 2 cancers-12-01018-f002:**
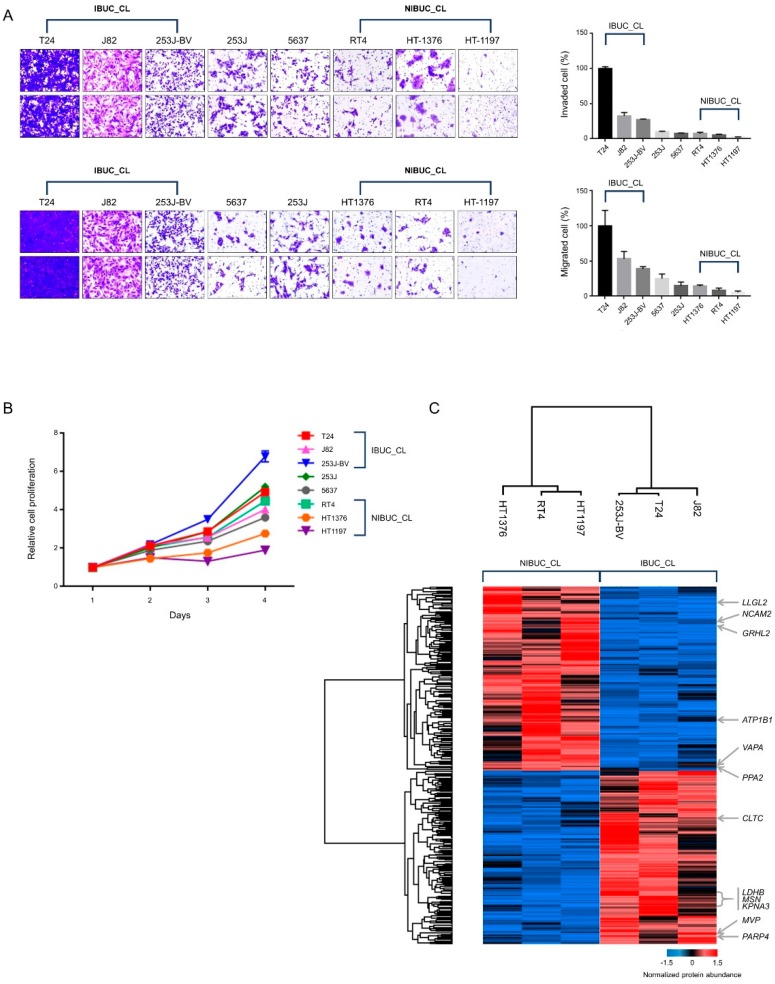
Bladder urothelial carcinoma (BUC) cell line results. (**A**) Invasion and migration Assay. (**B**) Proliferation assay. (**C**) Hierarchical clustering of differentially expressed proteins (DEPs) between invasive bladder urothelial carcinoma cell line (IBUC_CL) and non-invasive bladder urothelial carcinoma cell line (NIBUC_CL).

**Figure 3 cancers-12-01018-f003:**
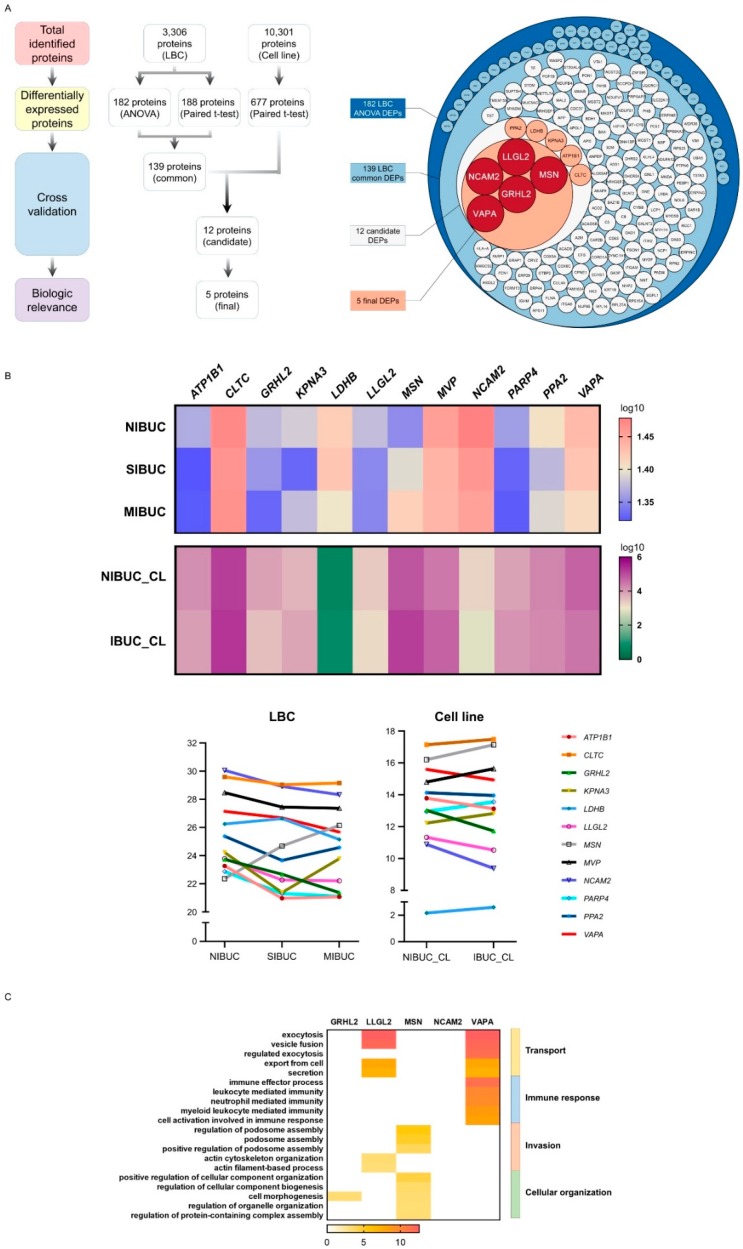
Invasion-associated biomarker selection. (**A**) Workflow for biomarker selection (left, overview; right, circlepack). (**B**) Intensity tendency of candidate biomarkers in BUC LBC and cell line proteomics (left, heatmap; right, broken-line graph). (**C**) Gene ontology results of final five candidate biomarkers.

**Figure 4 cancers-12-01018-f004:**
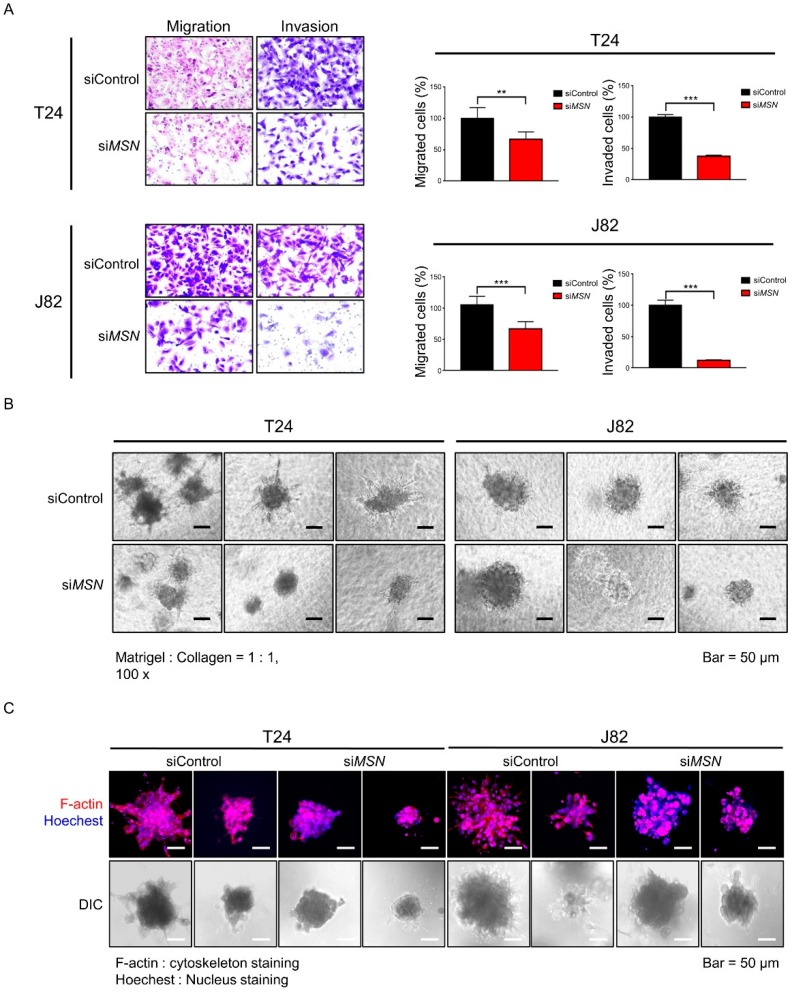
Functional validation of invasive role of *MSN* with small interfering RNAs (siRNAs) using two-dimensional (2D) and three-dimensional (3D) migration and invasion assays. (**A**) 2D migration and invasion assay (statistical significance, ** *p*-value < 0.01; *** *p*-value < 0.001). (**B**) Phase-contrast microscope image of 3D dissemination. (**C**) Confocal microscope image of 3D dissemination.

**Figure 5 cancers-12-01018-f005:**
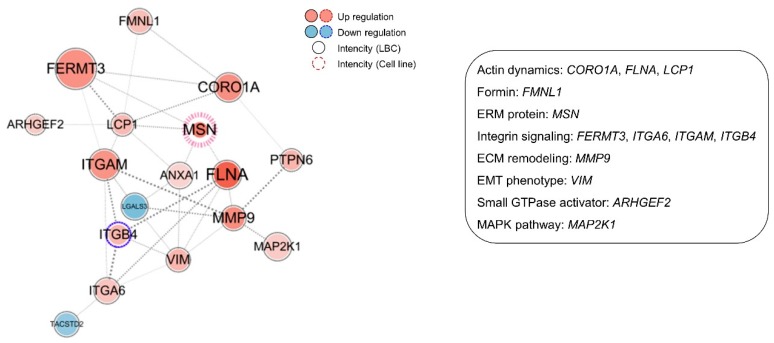
Protein-protein interaction based-network analysis with differentially expressed proteins (DEPs) between muscle-invasive bladder urothelial carcinoma (MIBUC) and non-invasive bladder urothelial carcinoma (NIBUC) in cell motility.

**Figure 6 cancers-12-01018-f006:**
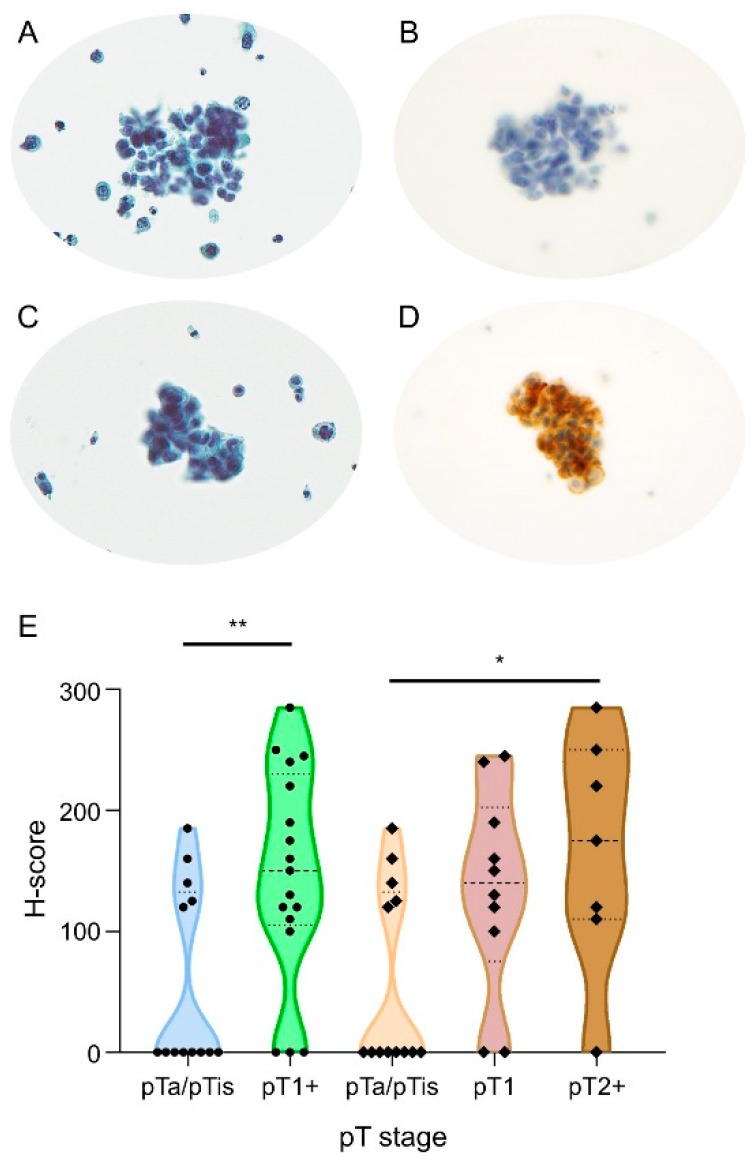
Immunocytochemical (ICC) verification of moesin (*MSN*) as a predictive biomarker for invasive bladder urothelial carcinoma (BUC) in liquid-based cytology (LBC) samples. (**A**–**D**) Representative BUC and moesin ICC images on LBC: (**A**) BUC LBC image, (**B**) matched ICC image (negative staining), (**C**) BUC LBC image, (**D**) matched ICC image (positive staining). (**E**) Moesin ICC positivity among non-invasive BUC (NIBUC) (pTa/pTis), stromal-invasive BUC (SIBUC) (pT1), and muscle-invasive BUC (MIBUC) (pT2+) (statistical significance, * *p*-value < 0.05; ** *p*-value < 0.01; pT, pathologic T; pTa, non-invasive papillary carcinoma; pTis, urothelial carcinoma in situ; pT1, tumor invades lamina propria (subepithelial connective tissue); pT2+, tumor invades muscularis propria and beyond).

**Table 1 cancers-12-01018-t001:** Correlation between moesin immunocytochemistry (ICC) and invasion depth of bladder urothelial carcinoma (BUC).

	Moesin Immunoreactivity, *n* (%)		*p*-Value	H-Score	*p*-Value
	Negative (*n* = 11)	Positive (*n* = 19)			
NIBUC (*n* = 13)	8 (72.7)	5 (26.3)	0.046	56.15	0.042
SIBUC (*n* = 10)	2 (18.2)	8 (42.1)		133.50	
MIBUC (*n* = 7)	1 (9.1)	6 (31.6)		165.71	
NIBUC (*n* = 13)	8 (72.7)	5 (26.3)	0.023	56.15	0.014
IBUC (*n* = 17)	3 (27.3)	14 (73.7)		146.76	

Abbreviations: IBUC, invasive bladder urothelial carcinoma; MIBUC, muscle-invasive bladder urothelial carcinoma; NIBUC, non-invasive bladder urothelial carcinoma; SIBUC, stromal invasive bladder urothelial carcinoma.

**Table 2 cancers-12-01018-t002:** Clinicopathologic features of bladder urothelial carcinoma (BUC) in liquid-based cytology (LBC) samples.

	Proteomic Analysis			ICC Validation		
	NIBUC, *n* (%)	SIBUC, *n* (%)	MIBUC, *n* (%)	NIBUC, *n* (%)	SIBUC, *n* (%)	MIBUC, *n* (%)
	*n* = 6	*n* = 5	*n* = 5	*n* = 13	*n* = 10	*n* = 7
Age (years)						
50-60	0 (0.0)	1 (20)	0 (0)	2 (15.4)	1 (10)	1 (14.3)
60-70	5 (83.3)	2 (40)	4 (80)	7 (53.8)	2 (20)	2 (28.6)
>70	1 (16.7)	2 (40)	1 (20)	4 (30.8)	7 (70)	4 (57.1)
Gender						
Male	5 (83.3)	5 (100)	5 (100)	12 (92.3)	10 (100)	6 (85.7)
Female	1 (16.7)	0 (0)	0 (0)	1 (7.7)	0 (0)	1 (14.3)
Pathologic diagnosis						
Suspicious for high-grade urothelial carcinoma	5 (83.3)	3 (60)	4 (80)	8 (61.5)	2 (20)	2 (28.6)
High-grade urothelial carcinoma	1 (16.7)	2 (40)	1 (20)	5 (38.5)	8 (80)	5 (71.4)
Papillary urothelial carcinoma, high-grade	6 (100)	4 (80)	0 (0)	13 (100)	10 (100)	3 (42.9)
Invasive urothelial carcinoma, high grade	0 (0)	1 (20)	5 (100)	0 (0)	0 (0)	4 (57.1)
Concurrent carcinoma in situ	1 (16.7)	2 (40)	1 (20)	5 (38.5)	3 (30)	0 (0)
pT stage						
pTa/pTis	6 (100)	0 (0)	0 (0)	13 (100)	0 (0)	0 (0)
pT1	0 (0)	5 (100)	0 (0)	0 (0)	10 (100)	0 (0)
pT2+	0 (0)	0 (0)	5 (100)	0 (0)	0 (0)	7 (100)
AJCC stage						
0a/0is	6 (100)	0 (0)	0 (0)	13 (100)	0 (0)	0 (0)
I	0 (0)	5 (100)	0 (0)	0 (0)	10 (100)	0 (0)
II+	0 (0)	0 (0)	5 (100)	0 (0)	0 (0)	7 (100)
Treatment						
Transurethral resection	6 (100)	5 (100)	1 (20)	13 (100)	10 (100)	1 (14.3)
Radical cystectomy	0 (0)	0 (0)	4 (80)	0 (0)	0 (0)	6 (85.7)

Abbreviations: AJCC, American Joint Committee on Cancer; ICC, immunocytochemistry; MIBUC, muscle-invasive bladder urothelial carcinoma, NIBUC, non-invasive bladder urothelial carcinoma; pT, pathologic T; pTa, non-invasive papillary carcinoma; pTis, urothelial carcinoma in situ; pT1, tumor invades lamina propria (subepithelial connective tissue); pT2+, tumor invades muscularis propria and beyond; SIBUC, stromal invasive bladder urothelial carcinoma.

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
