# Peer review of "Moesin (MSN) as a Novel Proteome-Based Diagnostic Marker for Early Detection of Invasive Bladder Urothelial Carcinoma in Liquid-Based Cytology"

_cancers, 2020, doi:10.3390/cancers12041018_

Round 1
Reviewer 1 Report
- How can the authors find DEPs in LBC and cell lines? Is the average amount of the same protein existing in each of 6 NIBUC specimens compared to those in SIBUC and MIBUC? Alternatively, the authors may use different methods for comparison.
- What are the standards for DEPs?
- The authors spend lots of time and effort to use LBC proteomics and cell line proteomics to find only five candidate biomarkers. Lastly, only MSN is chosen for further validation. However, only correlation is studied using a small cohort. The authors do not carry out univariable and multivariable analyses. It is not sure if MSN can become a novel biomarker for invasion. Can the authors discuss the above limitation?
Author Response
1. How can the authors find DEPs in LBC and cell lines? Is the average amount of the same protein existing in each of 6 NIBUC specimens compared to those in SIBUC and MIBUC? Alternatively, the authors may use different methods for comparison.
→ Thank you for your comments. We utilized Perseus bioinformatic software platform, which is the most widely used analytic tool developed by Matthias Mann group in Max Planck Institute [1]. As you commented, we individually performed statistical analyses for human LBC samples and urothelial carcinoma cell lines. For the LBC samples, we chose ANOVA option with 5% false discovery rate (FDR) level and filtered 182 DEPs among three groups. Therefore, NIBUC was compared to SIBUC and MIBUC, respectively, which was also applied to comparative analysis based on SIBUC or MIBUC. The cell lines samples were analyzed using Student t-test with 5% false discovery rate (FDR) level and showed 677 DEPs between two groups. As you mentioned, Perseus software has several normalization functions, which make average protein expression evenly distributed in each sample before statistical analysis.
2. What are the standards for DEPs?
→ We employed the statistical options, including ANOVA option with 1% false discovery rate (FDR) cut-off value and Student t-test with 1% false discovery rate (FDR) level for Student t-test for the more accurate selection of DEPs.
3. The authors spend lots of time and effort to use LBC proteomics and cell line proteomics to find only five candidate biomarkers. Lastly, only MSN is chosen for further validation. However, only correlation is studied using a small cohort. The authors do not carry out univariable and multivariable analyses. It is not sure if MSN can become a novel biomarker for invasion. Can the authors discuss the above limitation?
→ Our main purpose was to identify a biomarker for early detection of invasive bladder urothelial carcinoma. Also, we performed 2D and 3D invasion assay to determine whether the five candidate biomarkers are associated with bladder urothelial carcinoma invasion. Among five candidate biomarkers, only MSN revealed an association with invasion of bladder urothelial carcinoma cell lines. In cancer studies for proteomic biomarker discovery usually reported several biomarker candidiates and validated a few (<5) biomarkers according to their possible role in cancer development and progression [2,3]. We assessed immunocytochemistry for MSN to identify whether MSN has diagnostic significance in LBC slides, which are widely used for bladder urothelial carcinoma screening and follow-up. Our criteria for ICC validation cohort were 1) consistent cytologic diagnosis among three urologic pathologists (all three pathologists diagnosed SHGUC or HGUC or at least two of them diagnosed SHGUC or HGUC and other diagnosed atypical cells according to Paris system), 2) cellular specimen with fewer inflammatory cells (cellularity with > 50%), 3) no previous Bacillus Calmette–Guérin (BCG) treatment, 4) no neoadjuvant chemotherapy. Also, we tried to consist of validation cohort with similar epidemiologic samples. According to these criteria, we could gather relatively small 30 samples for ICC validation. On ICC validation we could identify that MSN immunoreactivity was positively associated with bladder urothelial carcinoma invasion. We did not perform clinicopathologic correlation study with bladder urothelial carcinoma tissue samples. However, in public TCGA data, we could identify that MSN expression is marginally associated with poor survival and angiolymphatic invasion. Also, in public TCGA data, MSN expression was associated with AJCC cancer staging. Based on these results, we suggested MSN as an invasion associated biomarker for bladder urothelial carcinoma. We thought further study for assessment of clinicopathologic correlation with MSN is needed on bladder urothelial carcinoma tissue samples. We added detailed criteria for ICC validation cohort in Supplementary methods (Page 4, Line 19-24).As the abbreviation BCG was added, we listed BCG in Abbreviation section of manuscript (Page 14, Line 355).

Reviewer 2 Report
The author tried to identify novel biomarkers to predict the tumor stages in urine cytology samples by using proteomics approach. As data showed that MSN protein could be a promising diagnostic marker for MIBUC. This study is interesting and provided a new strategy for treatment of IMBUC patients. Several minor issues were curiously.
- whether the author performed the animal model to demonstrate that MSN expression was involved in cancer metastasis.
- In figure 5, by GO analysis, MSN involved in many signaling pathways, such as actin dynamics, integrin signaling, ECM remodeling and MAPK signaling. These signals are important to regulate cell motility. Please confirm which signaling is crucial for MSN modulation. The author can examine the key proteins among these signalings in MSN-depleted cells.
Author Response
The author tried to identify novel biomarkers to predict the tumor stages in urine cytology samples by using proteomics approach. As data showed that MSN protein could be a promising diagnostic marker for MIBUC. This study is interesting and provided a new strategy for treatment of MIBUC patients. Several minor issues were curious.
1. whether the author performed the animal model to demonstrate that MSN expression was involved in cancer metastasis.
→ Thank you for your comments. We did not show the result of the animal model to evaluate whether MSN expression was involved in cancer metastasis. Alternatively, however, we performed 2D and 3D invasion assay using two bladder urothelial carcinoma cell lines. We thought further study to identify whether MSN expression was associated with cancer metastasis using animal models is encouraging and could suggest a more precise effect of MSN on cancer invasion and metastasis.
2. In figure 5, by GO analysis, MSN involved in many signaling pathways, such as actin dynamics, integrin signaling, ECM remodeling and MAPK signaling. These signals are important to regulate cell motility. Please confirm which signaling is crucial for MSN modulation. The author can examine the key proteins among these signalings in MSN-depleted cells.
→ As the reviewer commented, MSN was involved in many signaling pathways, including actin dynamics, integrin signaling, ECM remodeling, and MAPK signaling, which are essential to regulate cell motility. MSN is a member of ERM family and interacts with other proteins by binding its FERM and C-ERMAD domains. As the reviewer commented, proteomic evaluation of proteins illustrated in figure 5 would reveal the further pathogenic role of MSN. However, we did not quantify the aforementioned proteins by western blot and other methods, because we thought MSN interacts with other proteins by binding and may not affect the expression of those proteins in figure 5. Also, those proteins were not differentially expressed between invasive bladder urothelial carcinoma cell line (IBUC_CL) and non-invasive bladder urothelial carcinoma cell line (NIBUC_CL). Alternatively, we performed co-expression analysis using public TCGA data with MSN and key proteins in figure 5. As a result, we could find a strong correlation of MSN expression with actin dynamics (FLNA), integrin signaling (ITGAM), and EMT phenotype (VIM). These results suggest that MSN interacts with actin dynamics, integrin signaling, and EMT phenotype with the co-operative manner and is essential for cell motility. We added co-expression data (Figure S4) and mentioned this notion in the manuscript (Page 7, Line 168-171; Page 13, Line 312-315).
Figure S4. Co-expression analysis of MSN with key proteins in signaling pathway associated with cell motility. (A) Actin dynamics (B) Formin (C) Integrin signaling (D) Extracellular matrix (ECM) remodeling (E) Epithelial-mesenchymal transition (EMT) phenotype (F) Small GTPase activator (G) MAPK pathway (H) Others

Round 2
Reviewer 1 Report
No comments
Reviewer 2 Report
No.